# The Female-Specific W Chromosomes of Birds Have Conserved Gene Contents but Are Not Feminized

**DOI:** 10.3390/genes11101126

**Published:** 2020-09-25

**Authors:** Luohao Xu, Qi Zhou

**Affiliations:** 1Department of Neurosciences and Developmental Biology, University of Vienna, 1090 Vienna, Austria; luohao.xu@univie.ac.at; 2MOE Laboratory of Biosystems Homeostasis & Protection, Life Sciences Institute, Zhejiang University, Hangzhou 310058, China; 3Center for Reproductive Medicine, The 2nd Affiliated Hospital, School of Medicine, Zhejiang University, Hangzhou 310052, China

**Keywords:** sex chromosome, birds, sex-specific selection, gene expression

## Abstract

Sex chromosomes are unique genomic regions with sex-specific or sex-biased inherent patterns and are expected to be more frequently subject to sex-specific selection. Substantial knowledge on the evolutionary patterns of sex-linked genes have been gained from the studies on the male heterogametic systems (XY male, XX female), but the understanding of the role of sex-specific selection in the evolution of female-heterogametic sex chromosomes (ZW female, ZZ male) is limited. Here we collect the W-linked genes of 27 birds, covering the three major avian clades: Neoaves (songbirds), Galloanserae (chicken), and Palaeognathae (ratites and tinamous). We find that the avian W chromosomes exhibit very conserved gene content despite their independent evolution of recombination suppression. The retained W-linked genes have higher dosage-sensitive and higher expression level than the lost genes, suggesting the role of purifying selection in their retention. Moreover, they are not enriched in ancestrally female-biased genes, and have not acquired new ovary-biased expression patterns after becoming W-linked. They are broadly expressed across female tissues, and the expression profile of the W-linked genes in females is not deviated from that of the homologous Z-linked genes. Together, our new analyses suggest that female-specific positive selection on the avian W chromosomes is limited, and the gene content of the W chromosomes is mainly shaped by purifying selection.

## 1. Introduction

Sex chromosomes evolve in a distinctive manner from the rest of the genome, of which one (Y or W) chromosome is sex-limited except for the pseudoautosomal regions (PAR), while the other homologous (X or Z) chromosome is biasedly inherited in one of the sexes. Once the sex-linked regions cease homologous recombination on the Y or W chromosome in the heterogametic sex, they are usually subject to functional degeneration [1]. The degeneration process can be caused by various evolutionary mechanisms including genetic hitchhiking [2], Muller’s ratchet [3], and background selection [4]. Consequently, in many taxa with an ancestral origin of sex chromosomes, the male-specific Y (e.g., in eutherian mammals and *Drosophila*) [5,6] or female-specific W (e.g., in neognathous birds) [7,8] chromosomes have become highly heterochromatic and gene-poor, but the X and Z chromosomes remain euchromatic and gene-rich.

Due to the sex-limited or biased inheritance, sex chromosomes are hypothesized to be a preferred genomic location for the sexual antagonistic (SA) alleles to accumulate [9,10]. Since a Y-borne allele is only expressed in males, it is expected to accumulate male SA (beneficial to males, detrimental to females) alleles; likewise, female SA alleles are also expected to be preferentially located on the female-specific W chromosomes or mitochondrial genomes [11]. The accumulation of these SA alleles may accelerate the functional degeneration by the hitchhiking effect [12,13]. Studies have so far been focused on the male-specific Y chromosomes, and there are empirical data strongly supporting the ‘masculinization’ of the Y chromosomes due to sex-specific selection [1,14]. For instance, the Y chromosomes of mice have massively amplified testis genes [12]. On the *Drosophila* Y chromosomes, all of the genes are specifically expressed in testes [15,16,17], and there is a dramatic increase of copy numbers of testis genes on the recently evolved Y chromosome of *Drosophila* as well [18]. In both mammals [19] and *Drosophila* [20,21], such ampliconic genes can vary across closely related species, likely due to the independent amplification on the Y chromosomes.

On the contrary, the distribution of sex-related genes on the X chromosomes is more complicated. The X chromosomes are often demasculinized and feminized in many taxa, including the nematodes [22], *Drosophila* [23,24] and other Diptera species [25]. This can be caused by the fact that the X chromosome spends more time in females, therefore is expected to be an unfavorable location for genes of male functions but to be preferred by female-beneficial genes [26]. In addition, in organisms where the sex chromosomes are inactivated during meiosis, the demasculinization can be attributed to the evolutionary forces driving the testis-specific genes off the X chromosomes [27]. However, the genes expressed before the onset of meiotic sex chromosome inactivation can be overrepresented on the X chromosomes [19,28]. Moreover, recessive male-beneficial genes have a higher chance to accumulate on the X chromosomes that are favored by males [29,30].

Compared to the XY system, there are much fewer studies on whether and how sex-specific selections have shaped the gene content of the Z and W chromosomes. The Z chromosome of chicken has an excess of male-biased genes [31,32], and several amplicons have been identified at the end of the Z chromosome that appears to have testis-specific expression, presumably driven by male-biased selection [33]. It was suggested that the recombining part of the Z chromosome of emu has an excess of male-biased expression [34], but that is probably due to a lack of global dosage compensation in birds [35]. In contrast, chicken breeds that are selected for female fecundity have a higher W-linked gene expression than those of other breeds [36], conforming to the female-specific selection. An intensively studied W-linked gene *HINT1W* has been reported to have multiple copies in chicken [37,38] but has broad expression across many female tissues [39]. The copy number of *HINT1W* in songbirds, however, is only two or three [8], and is only one in duck (Li et al., personal communication). Those individual studies suggest while the Z chromosomes of birds seem to be masculinized, the evidence for the feminization of the W chromosomes is mixed.

To elucidate the evolutionary forces that impact the evolution of avian sex chromosomes, we collected the published female genomes from 11 songbirds [8] of Neoaves, one Galloanserae (chicken) [39], five ratites, and 10 tinamous [35,40] of Palaeognathae species that span the three major clades of birds. Particularly, we applied our analytical pipeline [8] to re-analyze the genomic data of Palaeognathae [40]. The comparative study of the sex-linked sequences from those species, combined with gene expression data from multiple tissues of both sexes, allows for a systematic assessment for the signature of W chromosome feminization. This study reveals deep conservation of gene repertoires of the avian W chromosomes, but finds little evidence of feminization of the W chromosome.

## 2. Materials and Methods

### 2.1. Genomic Dataset

The genomes and gametologous sequences of songbirds and chicken were directly retrieved from [8] and [39]. We re-analyzed the genomes of tinamous and ratites that were published in [40] and [35] using the same pipeline as in Xu el al. [8]. A full species list and their accession numbers can be found in the Appendix A.

### 2.2. Identification of the Sex-Linked Sequences in Paleognathae

We searched the homologous sex chromosome sequences in the newly assembled female genomes in Wang et al. [40] with a different pipeline used in Xu et al. [8]. To reduce both false positives and false negatives in the homologous searches, we chose the available Z chromosome sequence of most closely related species as a reference, rather than using the basal ostrich which is quite divergent from other paleognaths. For Nothurinae tinamous and Tinaminae tinamous, the reference Z chromosome sequences of Chilean tinamou and thicket tinamou were used, respectively, which have been identified from the male genomes [35]. Similarly, for emu/cassowary, rhea, kiwi, the reference Z chromosome of emu, greater rhea, and Okarito brown kiwi were used, respectively. We used nucmer (MUMmer 4.0.0.beta2) [41] to align the reference Z chromosome against the target female genomes, and only the 1-to-1 (delta-filter -1) best alignments were kept. When more than 60% of the sequence of a scaffold was aligned with reference Z chromosome, the scaffold was assigned as sex-chromosome linked. This threshold is different from the 50% of aligned sequences with at least 1 kb long, and sequence similarity between 70% and 95% used in [40].

To identify the W-linked sequences, we used a pipeline described in [8]: we first calculated the coverage of all scaffolds, and selected those that showed half the coverage level relative to autosomes. To do so, we aligned the female Illumina reads against the genomes using BWA-MEM (0.7.16a) with default parameters. The coverage at each nucleotide position was calculated with samtools (1.10) depth, but those with mapping quality lower than 60 (-Q 60) were removed. We then calculated the mean coverage over 50 kb sliding windows, but those windows with less than 30 kb sequences covered by reads were removed. We then removed those scaffolds that were Z-linked or autosomal according to their alignment against the reference male genomes. The retained candidate W-linked sequences were subject to individual inspections, i.e., whether they were homologous to the identified Z chromosomes.

### 2.3. Demarcation of the Paleognathous Evolutionary Strata

To study the distribution of Z/W similarity along the Z chromosomes, the pseudo-Z chromosomes (Z-linked scaffolds were ordered and joined into a chromosome) were reconstructed by Ragoo (1.1) [42] using the ostrich Z chromosome assembly from DNAZoo as the reference [43,44]. Before calculating the Z/W similarity, the repeat sequences were masked by RepeatMasker (4.0.7) using a repeat library combining bird repeats and paleognath-specific repeats. The W-linked sequences were aligned against the pseudo-Z chromosomes using LASTZ (1.04) with relaxed parameters “-step = 19 –hspthresh = 2200 –inner = 2000 –ydrop = 3400 –gappedthresh = 10,000”. Any alignment with very low aligned sequence percentage (<60%) or very short aligned size (<65 bp) was likely a false alignment and thus was filtered out. We also excluded alignments very high sequence identity (>96%) because they may represent alignments of repetitive sequences. The cleaned alignments were then concatenated by the order on the pseudo-Z chromosomes, and the Z/W sequence similarity was calculated by 100 kb non-overlapping windows.

To build the gametolog phylogenetic trees, we used the method described in Xu et al. (2019) [8]. The coding sequences of the Z-W gametologs were aligned using MAFFT (7.427) [45] with the “–auto” option. The alignments were filtered with trimAI (1.3) [46] to remove the gaps (-gt 0.2). We used IQ-TREE (2.0-rc1) [47] to construct the phylogenetic trees, with 100 times bootstrapping. The substitution models were automatically selected by IQ-TREE.

### 2.4. Gene Expression Analyses

The published RNA-seq data of chicken [48] and collared flycatchers [49] were downloaded from NCBI SRA. The raw reads were mapped using bowtie (1.2.1.1) [50] with default settings. The expression levels were estimated at the gene level using the RSEM (1.3.0) pipeline [51]. For each tissue, we calculated the mean expression levels of biological replicates. The expression data of green anole was retrieved from Xu et al. (2019) [8].

To identify the female-biased genes in green anole, we calculated the mean expression levels over replicates. We focused on the genes that are homologous to the bird sex-linked genes, and because they are autosomal in green anole, their expression represents the ancestral expression in the bird proto-sex chromosomes. Genes with zero or very low (TPM < 1) expression levels were removed from analyses. We calculated the female-to-male expression ratios, and used fold-change larger than 2 as a criterion to define female-biased genes. To assess the expression bias toward ovary relative to other female tissues, the ratio (R_ovary_) of expression of ovary (E_ovary_) over all tissues of males and females (brain, heart, kidney, liver, and gonad) was calculated, using the formula R_ovary_ = E_ovary_/sum(E_other_tissue_). If one gene had a ratio larger than 20%, it was defined as an ovary-dominant gene.

### 2.5. Gene Content Analysis

The chromosomal locations of the Z/W gene pairs were determined according to the location of their homologous genes on the Z chromosome of emu. For each gene where the emu–human orthologous relationship can be identified, the predicted haploinsufficiency score from human [52] was assigned to birds. Haploinsufficiency scores measure the level of haploinsufficiency which reflects the sufficiency a single copy of an allele to produce the standard phenotype. Although haploinsufficiency scores may change between species, this may not be the case of those genes with a high score. As a recent study shows that many haploinsufficient genes can be conserved across the animal kingdom even between human and yeast [53]. Based on the whole genome distribution of the haploinsufficency scores of all genes, we decided if the haploinsufficiency score is higher than 0.4, the gene was classified as dosage sensitive. If the mean expression level (TPM) is higher than 50, the gene was regarded as highly expressed.

### 2.6. HINT1W Copy Number Estimation

The copy number of HINT1W has been increased in many birds, potentially due to female-specific positive selection. To estimate the copy number, k-mer (k = 27) frequency was counted for the whole-genome sequencing reads using the count function of JELLYFISH (2.3.0) [54]. The haploid coverage was estimated to be half of the mean genomic coverage (at the peak of k-mer frequency distribution). We then generated the k-mers (k = 27) out of the coding sequencing of the assembled *HINT1W* using unikmer. For each of the *HINT1W*-derived k-mer, the count number was retrieved from the genomic k-mers using the query function of JELLYFISH. The coverage of *HINT1W* was then calculated by the median of the counts of *HINT1W*-derived k-mers.

## 3. Results

### 3.1. Independent Sex Chromosome Differentiation in Palaeognathous Birds

Sex chromosomes of birds share one time of recombination suppression, followed by multiple times of species or lineage-specific recombination suppression, forming a pattern called ‘evolutionary strata’. It is expected that the cease of recombination between the W and Z chromosome will lead to extensive sequence deletions, repeat accumulations, and gene loss on the W [7,8]. To investigate the extent of the W chromosome decay, first we demarcated the boundary of the pseudoautosomal regions (PARs) and the differentiation regions (DRs) according to the female coverage along the Z chromosome (Appendix A). Our re-analysis confirmed previous findings that kiwis and many tinamous have a pair of sex chromosomes with moderate differentiation, and one lineage of Tinaminae has a completely degenerated W chromosome (Figure 1a).

However, importantly, with a modified pipeline (Materials and Methods) our analysis retrieved a considerably large amount of W-linked sequences (Figure 1b) and W-linked genes (Figure 1c), with many of them derived from the oldest (S0) and second oldest (S1) strata of the sex chromosomes (Figure 1d, Appendix A). The more complete datasets of the W-linked sequences and genes of the S0 made it possible for us to better examine the conservation level of the W-linked S0 region that was previously considered highly degenerated across all birds. On the other hand, the independently evolved S1 of paleognaths and neognaths [7,35,40] is a useful model to identify common evolutionary forces that convergently shape the gene content of the avian W chromosomes.

### 3.2. Demarcation of the Evolutionary Strata of Sex Chromosomes

The wealth of the newly identified W-linked sequences is informative for the evolutionary history of sex chromosome differentiation across birds. The degree of sex chromosome differentiation is expected to be positively correlated with the age of recombination suppression, therefore the divergence of the homologous Z- and W-linked sequences can inform us of the age of evolutionary strata of the sex chromosome. The S0 evolved more than 102 million years ago [7], at the ancestor of all birds (Figure 2a), with the retained W-linked S0 showing a sequence similarity with the homologous Z at ≈70% (Figure 1d, Appendix A). The Z-W similarity is slightly higher in emu, cassowary, rhea, and ostrich than that in tinamous and kiwis (Appendix A), likely due to the lower mutation rates in those large-bodied ratites.

Previous work inferred that the S1 emerged independently in ostrich, emu/cassowary, rhea, kiwi, and tinamous [35,40], which is confirmed by our re-analyses (Figure 2a). For instance, the S1 of emu and cassowary is the smallest in size and also may be the youngest, shown by with the Z/W similarity level as high as 94% (Appendix A). Ostrich’s S1 is younger than that of greater rhea with a higher Z/W similarity level but with the same boundary (Appendix A); however, given their polyphyletic relationship, their S1 most likely evolved independently. The kiwi S1 is the largest among ratites, but evolved relatively recently, with the Z/W sequence similarity level at ≈90% (Appendix A). The S1 of tinamous is intermediate in size, emerging at the ancestor of tinamou at least 40 million years ago, consistent with its low Z/W similarity level (≈80%) (Figure 2a, Appendix A). To further validate the shared evolutionary history of S0 and the independent evolution of S1, we used the homologous Z/W gene pairs (gametologs) to reconstruct their phylogenetic relationships. If one evolutionary stratum evolved at the common ancestor of a clade, the gametologs would cluster by chromosome (Z or W) rather than by species. The phylogeny of *KCMF1* gametologs supports the single origin of S0 at bird ancestor (Figure 2b). Similarly, the S1-derived gene *ABHD17B* supports the independent origin of S1 in different paleognathous lineages (Figure 2c).

The third evolutionary stratum (S2) seems to have evolved only in elegant crested tinamou and at the ancestor of a Tinaminae group (Figure 2a, Appendix A). The S2 of elegant crested tinamous is evolutionarily young with the ZW sequence similarity level of ≈92%, while the S2 of the Tinaminae group is relatively older and larger, resulting in almost complete degeneration of the W chromosomes in this lineage (Figure 2a).

### 3.3. Deep Conservation of the Gene Content of the Avian W Chromosomes

To compare the sex-linked gene content across bird species, we next compiled a list of genes that survived on the degenerating W chromosomes at each evolutionary stratum. As among paleognaths the S2 is limited to a few tinamou species, our analysis only focused on the S0 and S1. In general, ratites have retained the largest number of genes on the W chromosomes, for both S0 and S1; tinamous retain a smaller number of W-linked genes, but larger than that of neognaths (Figure 3a,b). Thus, the rate of gene loss seems to be in concordance with the rate of sex chromosome differentiation among these species reported previously [8].

The second pattern that we found is that the retained genes are shared by different bird lineages (Figure 3a,b). If one gene is shared by at least two bird groups out of songbirds, chicken, ratites, and tinamous, it is regarded as a conserved gene; otherwise it is regarded as non-conserved. By this definition, for the W-linked genes of S0, 100%, ~89% and ~88% of them are conserved in chicken, songbirds and tinamous, respectively, but for ratites only ~43% are conserved (Figure 3a,c). Similarly, for S1 genes, 100%, 94% and ~63% are conserved in chicken, songbirds and tinamous respectively, and 79% are conserved in ratites (Figure 3b,d).

The conserved S0 genes seem to be randomly distributed along the Z chromosomes, with the most conserved gene *KCMF1* located in the middle (Figure 3a). Many of the conserved S1 genes, on the contrary, are located on the PARs of other species, but very close to the stratum boundaries. Specifically, *HNRNPK* and *UBQLN1* reside close to the S1 boundary of emu and cassowary, *SPIN1* is located at the S1 boundary of ostrich and rhea; and lastly, *CHD1*, frequently used as a bird sexing marker, is located close to the S1 boundary of tinamous (Figure 3b). Those five boundary genes are among the most conserved out of 132 gametolog-pairs (Figure 3e). The co-localization of the conserved genes and stratum boundaries raises the interesting possibility that a few critical genes may serve as a strong barrier of further sex chromosome differentiation in some paleognaths.

### 3.4. Genes on the W Chromosomes Have High Expression and High Dosage Sensitivity

The identification of genes that survive on the bird W chromosomes also allows us to compare the retained genes versus those that have become lost, in order to reveal the factors that dictate the retention of genes on the W chromosomes. Similar to previous reports (Bellott et al., 2017; Xu, Auer, et al., 2019), our analysis on the S1 and S0 shows that the retained genes, particularly the retained conserved genes, tend to have either higher expression levels or higher predicted dosage sensitivities, or both, relative to the lost genes (Figure 3e,f, Appendix A). Moreover, the dosage sensitivity is positively correlated with the conservation levels (percentage of presence among species) (*p* = 0.043) (Figure 3e). The disproportionate retention of dosage-sensitive and highly expressed genes suggests that purifying selection is an important force that impacts the retention of gametolog on the W chromosome. It is worth noting that the non-conserved S1 genes have a similar proportion of highly expressed genes or dosage-sensitive genes with the autosomal or lost genes (Figure 3f). This is likely because those non-conserved retained S1 genes reside in a younger stratum and may not have sufficient time to be deleted. After excluding the non-conserved S1, the overrepresentation of highly expressed or dosage-sensitive genes becomes statistically significant (Fisher’s exact test, *p* = 0.00037 and *p* = 0.010, respectively) (Appendix A) for the retained genes. However, a separate Fisher’s exact test for S0 or S1 alone did produce consistent significant results (Appendix A) likely because of the reduced power of statistics due to the much smaller gene numbers.

### 3.5. Little Evidence of Female-Specific Positive Selection on the Bird W Chromosomes

To examine the question of whether the W chromosomes of birds show signals of feminization, we ask whether the retained Z/W gene pairs were ancestrally female-biased and whether they have evolved female-biased expression since they became sex-linked. As the homologs of the bird sex-linked genes in the green anole lizard are autosomal, their expression levels can represent the ancestral gene expression on the proto-sex chromosome of birds. The genes retained on the bird W chromosomes do not seem to be more female-biased than those that were lost, in any of the four tissues investigated (Figure 4a). In particular, only one conserved W-linked gene has female-biased expression in one tissue (liver) (Figure 4a). Moreover, we did not detect an excess of ovary-dominant genes for the retained W-linked genes (Appendix A). Together, our analyses suggest genes that were selected to retain on the bird W chromosomes are probably not because of their pre-existing functional importance in females.

To investigate whether the retained genes on the W chromosome have evolved derived expression in female tissues, we examined the expression from four different tissues of both sexes of chicken and collared flycatcher for both the Z- and W-linked gametologs. The Z-linked gametologs are expressed in both sexes, with higher expression levels in males due to the lack of global dosage compensation, while the W-linked gametologs are exclusively expressed in females (Figure 4b). The expression profile of the W-linked gametologs is similar to that of the homologous Z-linked gametologs, without apparent changes after they become female-specific, and they tend to be expressed broadly (Figure 4b). Notably, though the W-linked gametologs are transcriptionally active, they have reduced their expression levels (Figure 4b), likely due to the global inactive chromatin status of the W chromosomes. Those patterns remain conserved across collared flycatcher and chicken.

The only duplicated or amplified gene that has been identified on the bird W chromosomes is *HINT1W* [8,55]. Despite the previous indication that this gene may be under female-specific selection [56], we did not detect a higher expression of this gene in any of the female tissues (Figure 4b). Moreover, it seems the increased copy number of *HINT1W* does not apply to all species. The copy-number of *HINT1W* is estimated to be three in North Island brown kiwi, but there is only a single copy in thicket tinamou (Appendix A). Moreover, apart from kiwis, the *HINT1* gene locus in other ratites resides in the PARs (Figure 3).

## 4. Discussion

The sex-linked region where recombination was suppressed in the ancestor of all birds (S0) emerged more than 102 million years ago. Over this long-term evolution of modern birds, a few genes are broadly retained across different bird clades on the gene-poor and highly heterochromatic W chromosomes. These genes are still actively and broadly transcribed across female tissues, though are downregulated compared with the homologous Z-linked gametologs. The expression of these genes may be critical for females to restore the imbalance of dosage due to sex chromosome differentiation. The W-linked gametologs’ role of restoring dosage balance has been discussed by studies in chicken [39] and songbirds [8], and this study demonstrated that the conclusion holds true when the birds of all major clades are considered.

The purifying selection against losing these conserved dosage-sensitive genes on the W chromosomes must be very strong so that independently evolved strata (S1) have convergently retained dosage-sensitive genes across bird taxa. The strength of purifying selection is in response to the slow rate of evolution of dosage compensation in birds [57]. Global dosage compensation is absent in birds [48,58,59], but gene-by-gene dosage compensation is reported through epigenetic regulation [60,61,62,63]. A recent study shows epigenetic dosage regulation has been established only at two loci on the chicken Z chromosome [64], one is close to the sex-determining region at S0 and the other at S1. Without rapid response to dosage imbalance, perhaps the most efficient way to maintain the dosage balance is to keep the genes from losing too soon. Due to the lower level of selective efficiency of the W chromosome, however, most of the less dosage-sensitive genes have been deleted. Intriguingly, a recent study suggested once deleted dosage-sensitive or housekeeping genes can be restored onto the W chromosomes through Z-linked transpositions [65].

The gene content of the avian W chromosomes has also been stable without frequent gene movement which is often seen in *Drosophila* and mammalian Y chromosomes. The Y-borne genes usually have testis-specific expression, frequently transposed from autosomes, as a consequence of male-specific selection. This does not seem to be the case for birds in which female-specific selection acting on the W chromosomes is weak, if any. It needs to be noted bird genomes in general have a much lower rate of gene movement [66], and retrotranspositions are rare [67,68]. Nevertheless, there has been a report that an autosomal gene (*NARF*) was retrotransposed into the W chromosome of the American crow [8], but it is unclear if this newly derived gene has evolved female-biased expression.

It is perhaps not surprising that female-specific selection on the W chromosome is weak, because sexual selection usually targets males. However, female-specific selection acting on the W chromosome is not completely absent. A recent study demonstrated that an ovary-biased gene (*ANAX1*) has been transposed onto the W chromosome of red bird-of-paradise [65]. In addition, in chicken breeds where female traits are selected, the W chromosome is upregulated due to female-specific selection [36]. However, the female-specific selection for female-beneficial loci is probably much weaker than the selection for maintaining dosage-sensitive genes.

Taken together, this study reveals the presence of purifying selection that shapes the gene content of the avian W chromosomes. The primary targets of selection are dosage-sensitive or housekeeping genes, only rarely female-beneficial genes. This is different from the mammalian and *Drosophila* sex chromosome systems which are male-heterogametic, but it needs to be taken into account that birds lack global dosage compensation which propels the need to maintain the W-linked copies of dosage-sensitive genes. Other female-heterogametic taxa in which dosage compensation is complete will be good models to test whether female-specific selection can be sufficiently strong to feminize the W chromosomes.

## Figures and Tables

**Figure 1 genes-11-01126-f001:**
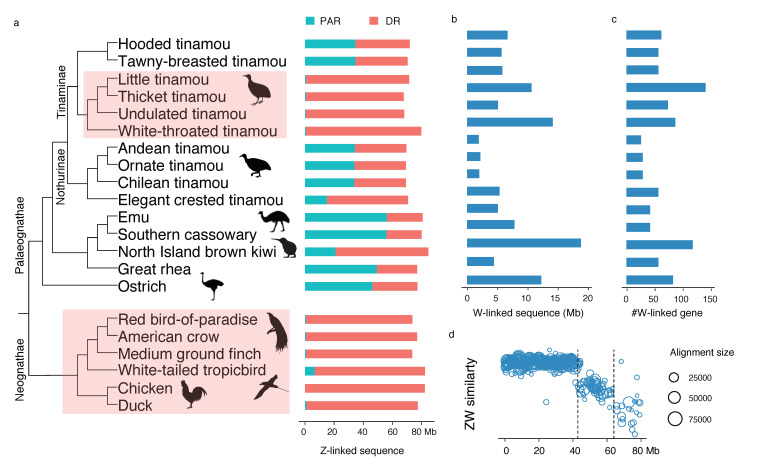
Identification of sex-linked sequences of birds. (**a**) The phylogeny contains the three major avian clades, showing the size of pseudoautosomal regions (PAR) and differentiated region (DR) of sex chromosomes. (**b**,**c**) This study retrieved W-linked sequences and genes for palaeognathous birds by re-analyzing the data from Wang et al. (2019). (**d**) The sequence similarity between the Z and W chromosomes of the white-throated tinamou can be divided (by the dashed lines) into three evolutionary strata. Each dot represents a 100-kb window. The *X*-axis represents the positions on the Z chromosomes.

**Figure 2 genes-11-01126-f002:**
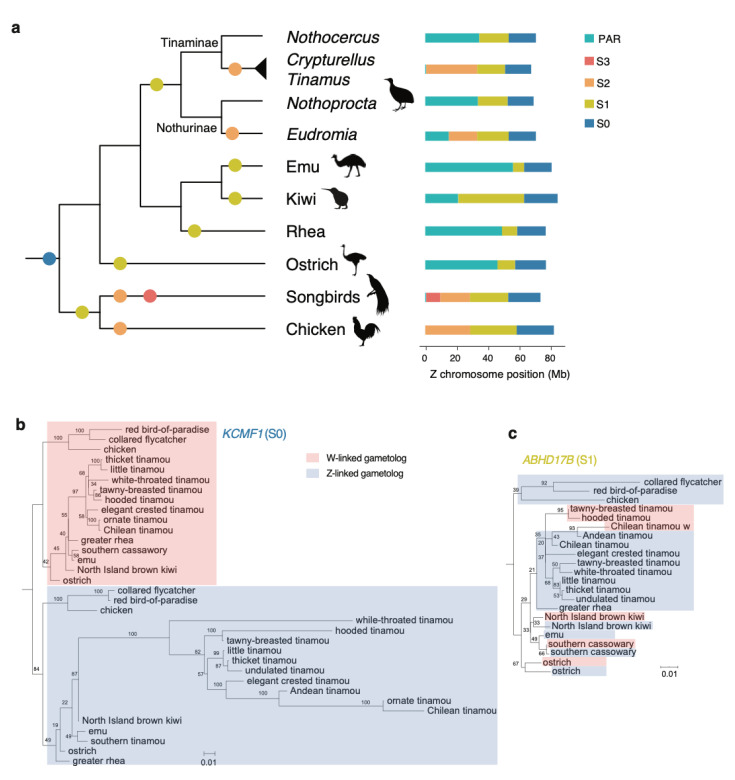
Independent evolutionary strata of sex chromosomes in Palaeognathae. (**a**) The differentiated region of sex chromosomes was divided into evolutionary strata according to the divergence level between the Z and W chromosomes. The colored dots along the branches of the phylogeny indicate the origins of evolutionary strata. The gametolog tree of *KCMF1* in (**b**) supports a single origin of S0 in birds, and the gametolog tree of *ABHD17B* in (**c**) supports multiple independent origins of S1 in Palaeognathae. The numbers at the phylogenetic nodes indicate the bootstrapping values.

**Figure 3 genes-11-01126-f003:**
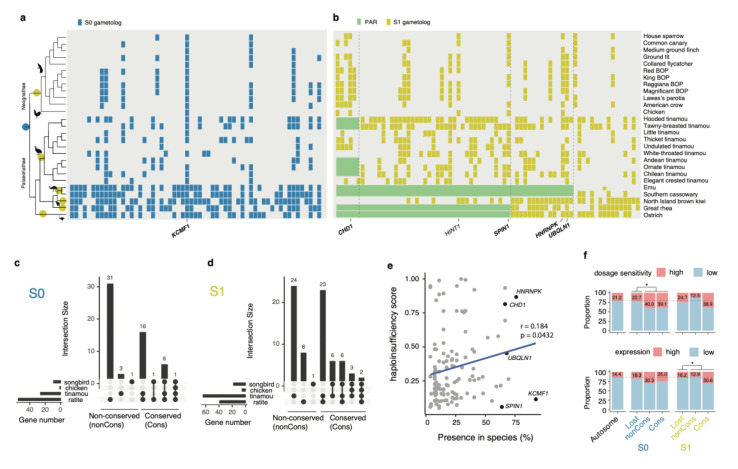
Conservation of W chromosome gene content maintained by purifying selection. (**a**,**b**) The colored circles on the phylogenetic branches indicate the origins of S0 (blue) and S1 (yellow). Each tile represents a W-linked gametolog. The genes are ordered according to their homologous positions on the emu Z chromosome. Five conserved boundary genes and *HINT1* are highlighted. (**c**,**d**) The UpSet plots show the numbers of shared W-linked gametologs across three groups of songbirds, chicken, tinamous, and ratites. When one gene was shared by at least two groups of birds, it was defined as conserved. (**e**) The *X*-axis represents the percentage of species where the W-linked genes are present. It is positively correlated with the gene’s predicted haploinsufficiency score of human ortholog (Pearson’s correlation coefficient and the *p*-value are shown). Black dots represent the five conserved genes at the strata boundaries. (**f**) Genes were defined as having high dosage sensitivity and high expression level when the predicted haploinsufficiency scores are larger than 0.4 and expression level (TPM) are larger than 50, respectively. The dotted line shows the average levels of autosomes. The asterisks represent significant results (*p*-value < 0.05) of Fisher’s exact test.

**Figure 4 genes-11-01126-f004:**
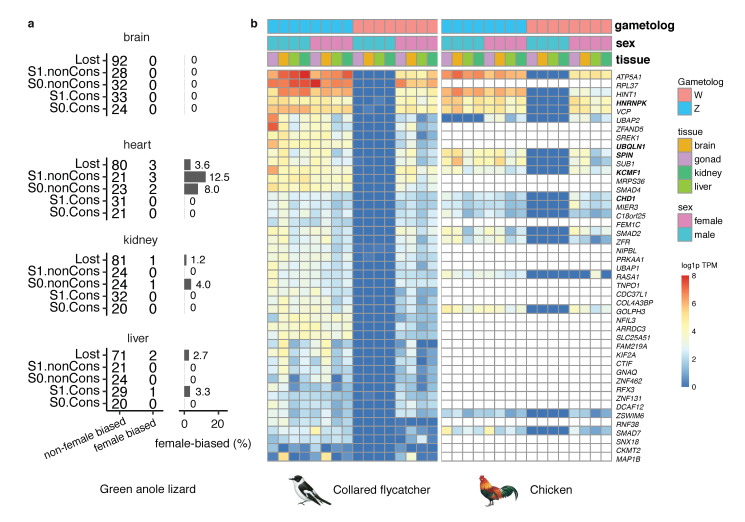
There is little evidence of feminization on the avian W chromosome. (**a**) The left panel shows the numbers of female-biased and non-female-biased genes in four tissues of the green anole lizard. The right panel shows the ratio of female-biased genes for each gene category. (**b**) The expression levels of genes that are present on the W chromosomes of collared flycatcher and chicken. Each row represents the expression level of both and Z- and W-linked gametologs in male and female tissues. The expression profile of W-linked gametologs in females is similar to that of the homologous Z-linked gametolog. Five conserved boundary genes are shown in bold. Log1p(TPM) means log(1 + TPM value).

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
