# Peer review of "The Female-Specific W Chromosomes of Birds Have Conserved Gene Contents but Are Not Feminized"

_genes, 2020, doi:10.3390/genes11101126_

Round 1

Reviewer 1 Report

This manuscript reports an comparative genomic analysis of sex chromosome evolution across a diversity of avian taxa, emphasizing the gene content and relative conversation of the degenerate W chromosome. While this is a re-analysis of previously existing data, the application of additional bioinformatic efforts to identify the W-linked portions of the genome assemblies has yielded substantially more W-linked sequence than prior studies of the same data, particularly for the oldest S0 stratum. Notable results include the various independent origins of S1, the lack of strong feminized expression in the W gametologs, and pattern of increased haploinsufficiency among conserved W gametologs.

Overall I view this study as well executed and the manuscript as well-written with lucid theoretical contextualization of the motivation and results.  I do not note any major flaws or shortcomings in the study. My critiques here primarily revolve around minor errors in the text, and also a request for additional methodological detail at some points.

The one peculiarity I note about this publication is that to a very large extent, it depends on data and even results from the Wang et al. biorxiv preprint, which seems to come from the same lab (Senior Author is also Zhou). It seems odd for a lab to publish this new paper, contradicting a previous finding from the same lab (Wang et al) that is apparently not yet in press.  Perhaps it is forthcoming and the timelines will line up. But it would seem reasonable to suggest that publication of the current study should be contingent on proper publication of the Wang paper.

Other comments are minor, as follows:

Section 2.2:

“To identify the W-linked sequences, we first calculated the coverage of all scaffolds, and selected those that showed half the coverage level relative to autosomes."

Some more detail here is warranted concerning methods and metrics for calculating coverage and performing Illumina alignments. What software was used for read alignments? Were you calculating “coverage” counts of reads or average depth across nucleotides? Also, it does not say explicitly but this must be female Illumina data, based on the logic.

Section 2.5: “For each gene where the emu-human orthologous relationship can be identified, the predicted haploinsufficiency score from human was assigned to birds. If the haploinsufficiency score is higher than 0.4, the gene was classified as dosage sensitive.”

What is the source of haploinsufficiency data? Which database or publication?  Also, what is the justification for using 0.4 as the threshold? This seems arbitrary without further explanation. Why not a different value?

Section 3.1:  TYPO:  differentation -> differentiation

Section 3.2:  “The S1 emerged independently in ostrich, emu/cassowary, rhea, kiwi and tinamous, with different sizes and ages in different lineages (Figure 2a).”

This paragraph fails to communicate the rationale of the inference as effectively as it could. This is because it makes a  strong claim about evolutionary history and then follows with a series of statements about the different lineages, but fails to explain the logic by which the original claim is supported. It feels like a litany of facts that the reader is left alone to puzzle out the logic that supports the claim. Please revise to do a better explaining how the facts that follow combine to support the notion of independent S1 evolution.

For the phylogenies in Fig 2, what methods/software were used to build them? Probably they are maxlikelihood based on nucleotide data, but I shouldn’t have to be guessing at this.

On pg 6, when discussing the “other” Chilean tinamou data, you do not explicitly reference the Xu paper. Thus it seems that you may have generated these data yourself. Please revise to be more explicit about where the alternative data came from.

First line of discussion:  “is common all birds”   should be “common _to_ all birds”.

First paragraph of discussion:  “transcribed in broad tissues” is a phrase that makes little sense. As written, it means the tissues themselves are broad. I believe you mean that the expression is broad across tissues, or in other words, broadly transcribed.

The legend of figure 4 indicating expression level is labelled “(log 1p TPM)”, which I think is probably a typo.  Maybe it is log10?

Fig S3 legend: Please carefully proofread the legend.  For instance, should be “previously” and there are one too many “is” in a sentence.

Author Response

Review 1 (revisions highlighted in green)

This manuscript reports an comparative genomic analysis of sex chromosome evolution across a diversity of avian taxa, emphasizing the gene content and relative conversation of the degenerate W chromosome. While this is a re-analysis of previously existing data, the application of additional bioinformatic efforts to identify the W-linked portions of the genome assemblies has yielded substantially more W-linked sequence than prior studies of the same data, particularly for the oldest S0 stratum. Notable results include the various independent origins of S1, the lack of strong feminized expression in the W gametologs, and pattern of increased haploinsufficiency among conserved W gametologs.

Overall I view this study as well executed and the manuscript as well-written with lucid theoretical contextualization of the motivation and results.  I do not note any major flaws or shortcomings in the study. My critiques here primarily revolve around minor errors in the text, and also a request for additional methodological detail at some points.

The one peculiarity I note about this publication is that to a very large extent, it depends on data and even results from the Wang et al. biorxiv preprint, which seems to come from the same lab (Senior Author is also Zhou). It seems odd for a lab to publish this new paper, contradicting a previous finding from the same lab (Wang et al) that is apparently not yet in press.  Perhaps it is forthcoming and the timelines will line up. But it would seem reasonable to suggest that publication of the current study should be contingent on proper publication of the Wang paper.

R: The first author Luohao Xu was not involved in Wang et al.’s work, and after reading the submitted manuscript posted on bioRxiv, he re-analysed the data with a new pipeline, and slightly different datasets (e.g. two tinamou individuals rather than one individual in Wang et al.) , and in a different phylogenetic scope. Wang et al. focused on sex chromosome evolution history of Palaeognathae, while this paper focused on a broader question: how conserved are the W-linked genes throughout the entire avian clade? We believe this manuscript positively complements Wang et al.’s work. When we submitted this manuscript to Genes, the Wang et al. paper was already in the process of second revision without major changes needed. We will discuss with the editor, if this manuscript is accepted by Genes, it can be only published after the publication of Wang et al. paper.

Other comments are minor, as follows:

Section 2.2:

“To identify the W-linked sequences, we first calculated the coverage of all scaffolds, and selected those that showed half the coverage level relative to autosomes."

Some more detail here is warranted concerning methods and metrics for calculating coverage and performing Illumina alignments. What software was used for read alignments? Were you calculating “coverage” counts of reads or average depth across nucleotides? Also, it does not say explicitly but this must be female Illumina data, based on the logic.

R: We have added more details at L119-125

Section 2.5: “For each gene where the emu-human orthologous relationship can be identified, the predicted haploinsufficiency score from human was assigned to birds. If the haploinsufficiency score is higher than 0.4, the gene was classified as dosage sensitive.”

What is the source of haploinsufficiency data? Which database or publication?  Also, what is the justification for using 0.4 as the threshold? This seems arbitrary without further explanation. Why not a different value?

R: We have added the citation Huang et al. 2010. We added in the text we decided the threshold of 0.4 according to the whole-genome distribution pattern.

Section 3.1:  TYPO:  differentation -> differentiation

R: Corrected at L187

Section 3.2:  “The S1 emerged independently in ostrich, emu/cassowary, rhea, kiwi and tinamous, with different sizes and ages in different lineages (Figure 2a).”

This paragraph fails to communicate the rationale of the inference as effectively as it could. This is because it makes a  strong claim about evolutionary history and then follows with a series of statements about the different lineages, but fails to explain the logic by which the original claim is supported. It feels like a litany of facts that the reader is left alone to puzzle out the logic that supports the claim. Please revise to do a better explaining how the facts that follow combine to support the notion of independent S1 evolution.

R: We have revised this part according to the reviewer’s suggestions at L228.

For the phylogenies in Fig 2, what methods/software were used to build them? Probably they are maxlikelihood based on nucleotide data, but I shouldn’t have to be guessing at this.

R: we have added a paragraph (L143) to describe the phylogenetic method. We thank the reviewer for pointing out this missing part.

On pg 6, when discussing the “other” Chilean tinamou data, you do not explicitly reference the Xu paper. Thus it seems that you may have generated these data yourself. Please revise to be more explicit about where the alternative data came from.

R: We have added the citation (Xu et al. 2019 GBE).

First line of discussion:  “is common all birds”   should be “common _to_ all birds”.

R: Corrected at L362.

First paragraph of discussion:  “transcribed in broad tissues” is a phrase that makes little sense. As written, it means the tissues themselves are broad. I believe you mean that the expression is broad across tissues, or in other words, broadly transcribed.

R: The reviewer is right, we have revised it accordingly at L365.

The legend of figure 4 indicating expression level is labelled “(log 1p TPM)”, which I think is probably a typo.  Maybe it is log10?

R: This is not actually a typo, ‘log1p() returns log(1 + number ) computed in a way that is accurate even when the value of the number is close to zero’. We added a note into the Figure 4 legend.

Fig S3 legend: Please carefully proofread the legend.  For instance, should be “previously” and there are one too many “is” in a sentence.

R: We have corrected the typos in the legend.

Reviewer 2 Report

The authors ask an important and interesting question - What is the role of sex-specific selection in the evolution of female-heterogametic sex chromosomes? The number and diversity of methods used is impressive. However, I think the manuscript can be improved by 1) providing justification for each of the methods, and 2) Clarifying the relationships between gene expression/dosage sensitivity/haploinsufficiency and selection. Some of the revisions could be made simply through rearrangement of text (moving justification sentences from the results to the methods) while other revisions will require a little more work.

Major Revisions

  1. The author’s main conclusion - That female-specific selection in birds (with heterogametic sex chromosomes) is limited - is not directly supported by their data. The authors could use a simple test of selection, such as dN/dS, to support or refute this claim. Additionally, the authors currently do not provide an explicit connection between expression or dosage sensitivity to selection.
  2. Overall, the justification for each of the methods need to be explicitly stated and/or clarified. It is difficult for me to evaluate the validity of the methods/assumptions when the justification is not explicitly stated. The authors should provide these justification statements at the beginning of each methods section (not the results section). The authors need to provide justification for:
  1. Reanalysing data from Wang 2019 - Why do the Paleognaths receive this extra analysis, but the other bird groups do not? Why is a 60% threshold better than the 70 - 95% used in Wang’s study?
  2. Demarcation of paleognathous evolutionary strata - Authors should define pseudo-Z chromosomes and justify excluding very high sequence identity (>96 %) on the basis of likely being a false alignment.
  3. Gene expression analyses - Why were female-biased genes in green anole identified? (This is somewhat explained in the results section 3.5, but it should be in the methods, and greater explanation on why this may represent an ancestral autosomal gene expression pattern).
  4. Sex expression bias - clarify what the ratio is (expression ovary/brain+heart+kidney…. OR ovary/brain + ovary/heart….. OR something else)
  5. Gene content analysis - How would associating haploinsufficiency with # of species support or refute their hypothesis? What are the potential pitfalls of using a human haploinsufficiency index?
  6. HINT1W copy number estimation - Why is assessing the numbers of HINT1W copies important?

3) Phylogenetic methods need to be provided for figure 2. 

Minor Revisions

Provide a citation for the phylogenetic tree you are using in Figure 1 - if it is a phylogeny you constructed, please provide information on how you constructed it.

Define Pseudo-Z chromosome

Define haploinsufficiency score

Section 3.2 second paragraph - there should be a space between Ostrich’s and S1

Figure 4b - Reorder the top 3 rows (1 gametolog, 2 tissue, 3 sex) to (1 gametolog, 2 sex, 3 tissue) and provide more information as to what each column is this figure was very hard to interpret and I was unable to figure out exactly what I was supposed to be taking away from it.

Author Response

Review 2 (revisions highlighted in yellow)

The authors ask an important and interesting question - What is the role of sex-specific selection in the evolution of female-heterogametic sex chromosomes? The number and diversity of methods used is impressive. However, I think the manuscript can be improved by 1) providing justification for each of the methods, and 2) Clarifying the relationships between gene expression/dosage sensitivity/haploinsufficiency and selection. Some of the revisions could be made simply through rearrangement of text (moving justification sentences from the results to the methods) while other revisions will require a little more work.

Major Revisions

  1. The author’s main conclusion - That female-specific selection in birds (with heterogametic sex chromosomes) is limited - is not directly supported by their data. The authors could use a simple test of selection, such as dN/dS, to support or refute this claim. Additionally, the authors currently do not provide an explicit connection between expression or dosage sensitivity to selection.

R: In this work we tested two forms of selection, one is female-specific positive selection and the other is purifying selection, that may play a role in shaping the gene content of the avian W chromosomes. Previous studies have clearly shown that purifying selection drives the survival of dosage-sensitive genes or housekeeping genes on songbird and chicken W chromosomes (Bellot et al. 2017; Xu et al. 2019). In this study, we demonstrated that purifying selection applies to all birds. To make the connection between purifying selection and expression or dosage sensitive more explicit, we added “The disproportionate retention of dosage-sensitive and highly expressed genes suggests purifying selection is an important force that impacts the retention of gametolog on the W chromosome” at section 3.4.

Female-specific positive selection is more difficult to test in non-model organisms without population genetic data. While the dN/dS statistic is useful to test positive selection between species, as suggested by the reviewer, it does not reflect the female-specific functions of the tested genes. Therefore in this study we test the female-specific selection by asking whether the expression of the retained W-limited genes is ancestrally female-biased or whether they have evolved biased expression in female organs. Since we did not detect female-biased expression of the W-linked genes, we reasoned it’s likely female-biased positive selection is not playing an important role, at least not the major force that governs the retention of genes on the W chromosomes, and have not driven the W chromosome feminized. A similar method of using gene expression data to infer possible positive selection on the Y chromosomes has been applied in Drosophila (Zhou and Bachtrog 2012) and mouse (Soh et al. 2014)

  1. Overall, the justification for each of the methods need to be explicitly stated and/or clarified. It is difficult for me to evaluate the validity of the methods/assumptions when the justification is not explicitly stated. The authors should provide these justification statements at the beginning of each methods section (not the results section). The authors need to provide justification for:
  1. Reanalysing data from Wang 2019 - Why do the Paleognaths receive this extra analysis, but the other bird groups do not? Why is a 60% threshold better than the 70 - 95% used in Wang’s study?

R: We have updated the text at L93. We speculate Wang et al are too conservative in their estimation of W-linked genes, and some genes may be missing because they used a somewhat distant reference genome. Therefore we decided to re-analyse the Paleognaths data. Because the other birds like songbirds (Xu et al. 2019) were analyzed by the same pipeline, we did not reanalyze them. We did not require the sequence similarity threshold, rather the 60% stands for percentage of the aligned sequence, not sequence similarity. We clarified this in the method at L119 and 142.

  1. Demarcation of paleognathous evolutionary strata - Authors should define pseudo-Z chromosomes and justify excluding very high sequence identity (>96 %) on the basis of likely being a false alignment.

R: We have updated the text at L134-144

  1. Gene expression analyses - Why were female-biased genes in green anole identified? (This is somewhat explained in the results section 3.5, but it should be in the methods, and greater explanation on why this may represent an ancestral autosomal gene expression pattern).

R: We added “We focused on the genes that are homologous to the bird sex-linked genes, and because they are autosomal in green anole, their expression represents the ancestral expression in the bird proto-sex chromosomes” at L142-145.

  1. Sex expression bias - clarify what the ratio is (expression ovary/brain+heart+kidney.... OR ovary/brain + ovary/heart..... OR something else)

R: The text was updated at L161-163.

  1. Gene content analysis - How would associating haploinsufficiency with # of species support or refute their hypothesis? What are the potential pitfalls of using a human haploinsufficiency index?

R: When a gene has a higher haploinsufficiency score (higher dosage-sensitivity), the purifying selection maintaining it on the W chromosome is stronger, which results in its retention in more species. It’s possible the haploinsufficiency scores may have turnovers between human and birds, however, for haploinsufficient genes the scores are likely conserved because they are likely involved in complex networks and housekeeping genes. Because to some extent the haploinsufficient genes are conserved between human and S. cerevisiae (Clare et al. 2011 BMC Biology), we have reasons to believe they should also be conserved between human and birds. We have clarified this at L175-178.

  1. HINT1W copy number estimation - Why is assessing the numbers of HINT1W copies important?

R: We updated the text at Line 184.

3) Phylogenetic methods need to be provided for figure 2.

R: We have added one paragraph in the section 2.3 describing the methods used for phylogenetic construction at L147.

Minor Revisions

Provide a citation for the phylogenetic tree you are using in Figure 1 - if it is a phylogeny you constructed, please provide information on how you constructed it.

R: We have added citations in the Figure 1 legend at L218

Define Pseudo-Z chromosome

R: in the section 2.3 we added: “Z-linked scaffolds were ordered and joined into a chromosome” in the bracket right after pseudo-Z chromosome

Define haploinsufficiency score

R: We added the definition in the section 2.5: Haploinsufficiency scores measure the level of haploinsufficiency which reflects the sufficiency of a single copy of an allele to produce the standard phenotype.

Section 3.2 second paragraph - there should be a space between Ostrich’s and S1

R: We have added the space

Figure 4b - Reorder the top 3 rows (1 gametolog, 2 tissue, 3 sex) to (1 gametolog, 2 sex, 3 tissue) and provide more information as to what each column is this figure was very hard to interpret and I was unable to figure out exactly what I was supposed to be taking away from It.

R: We have reordered the top 3 rows as suggested by the reviewer. We have also added in the legend “Each row represents the expression level of both and Z- and W-linked gametologs in male and female tissues. The expression profile of W-linked gametologs in females is similar to that of the homologous Z-linked gametolog”.

Reviewer 3 Report

In this study, Xu and Zhou used a new pipleine combined with multiple expression data to elucidate the evolutionary forces that impact the sex chromosome evolution in the three three major avian clade, Neoaves, Galloanserae and Palaeognathae. The pipleine that they generated will be broadly useful for the community as it identifies genes of interest for future study. The methodology and the details of the generated data are clearly described and appropriate. The pipeline allowed the ientification of more W-linked genes within Palaeognathae than a previous study. The authors found that the avian W chromosomes have conserved gene content, but found little evidence of feminization of the W chromosome. These observations provide information about the gene content and dynamics of sex-chromosome gene expression in birds, but I think that the authors should provide more information about how the pylogenies were genereated before publication. Figure 1 have 3 phylogenetic trees but authots did not mention how these trees were retrieved.

Author Response

In this study, Xu and Zhou used a new pipleine combined with multiple expression data to elucidate the evolutionary forces that impact the sex chromosome evolution in the three three major avian clade, Neoaves, Galloanserae and Palaeognathae. The pipleine that they generated will be broadly useful for the community as it identifies genes of interest for future study. The methodology and the details of the generated data are clearly described and appropriate. The pipeline allowed the ientification of more W-linked genes within Palaeognathae than a previous study. The authors found that the avian W chromosomes have conserved gene content, but found little evidence of feminization of the W chromosome. These observations provide information about the gene content and dynamics of sex-chromosome gene expression in birds, but I think that the authors should provide more information about how the pylogenies were genereated before publication. Figure 1 have 3 phylogenetic trees but authots did not mention how these trees were retrieved.

R: We have added one paragraph in the section 2.3 describing the methods used for phylogenetic construction.